# Investigating Strategies of Emotion Regulation As Mediators of Occupational Stressors and Mental Health Outcomes in First Responders

**DOI:** 10.3390/ijerph19127009

**Published:** 2022-06-08

**Authors:** Sowmya Kshtriya, Jacqueline Lawrence, Holly M. Kobezak, Paula J. Popok, Sarah Lowe

**Affiliations:** 1Department of Psychology, Montclair State University, Montclair, NJ 07043, USA; jacquelinen.lawrence@gmail.com; 2Department of Psychology, Binghamton University, Binghamton, NY 13902, USA; hkobezak96@gmail.com; 3Department of Psychiatry, Massachusetts General Hospital, Boston, MA 02114, USA; paula.popok@gmail.com; 4Department of Social and Behavioral Sciences, Yale School of Public Health, New Haven, CT 06510, USA; sarah.lowe@yale.edu

**Keywords:** first responders, emotion regulation, expressive suppression, cognitive reappraisal

## Abstract

The aim of this study was to investigate whether two emotion regulation strategies, expressive suppression or cognitive reappraisal, mediated the development of posttraumatic stress disorder (PTSD), major depression (MD), and generalized anxiety disorder (GAD) in first responders (FR) who experienced occupational stressors, using cross-sectional data. An aggregate of 895 first responders (M = 37.32, SD = 12.09, 59.2% male, 91.3% Caucasian) who were recruited through professional organizations and social media sites across North and South American states participated in an online Qualtrics survey. Bivariate correlation analyses demonstrated that occupational stressors were positively correlated with expressive suppression and each mental health outcome but were not significantly correlated with cognitive reappraisal. Mediation analyses demonstrated expressive suppression as a significant mediator between occupational stressors and PTSD, MD, and GAD symptoms, but not cognitive reappraisal. Even though these findings are in purview of a cross-sectional research design, they suggest the importance of practices that bolster first responders’ ability to use more effective and adaptive emotion regulation strategies such as emotion expression, effective communication, and cognitive reappraisal that might help enhance psychological resilience.

## 1. Introduction

First responders (FR) are emergency workers who are responsible for protecting the community and environment during the early stages of a critical incident. They include firefighters, paramedics, hospital staff, and rescue personnel [1,2,3]. Due to the nature of their jobs, they often face multifarious work-related stressors, also known as occupational stressors, including a lack of sleep, fear of safety, overwhelming workloads, or overexertion [4]. Since work-related stressors are, to some degree, an inherent part of first responders’ jobs, it is important to identify factors that account for the relationship between stressors and mental health.

Knowing what factors lie along the pathway from work-related stressors to mental health could be useful in identifying strategies to mitigate adverse psychological impacts. One potential factor along the path from occupational stressors is emotion regulation. Emotion regulation includes automatic (unconscious) and controlled (conscious) processes that are used to modulate an emotional response [5,6], and it is the capacity by which individuals perceive their experience of emotions, influence their emotions, and express them accordingly [7,8]. Prior research has shown that, because the occupational stressors that first responders face on a daily basis require a great deal of emotion regulation or emotion response strategies, first responders often face increased risk for emotionally laden mental health problems, including symptoms of generalized anxiety disorder (GAD), major depression (MD), and posttraumatic stress disorder (PTSD) [1,4,9].

Two commonly used emotion regulation strategies are expressive suppression and cognitive reappraisal [7,10]. Expressive suppression is when an individual conceals, inhibits, or suppresses feelings and corresponding emotion-expressive behaviors. Cognitive reappraisal is when an individual reframes or reinterprets the emotionally triggering situation in order to alter its meaning and impact on the emotion [5,6,7,10]. According to Gross’s process model of emotion regulation [5,7], expressive suppression is categorized as a “response-focused” emotional regulation strategy, where the emotional stimuli are modified *during or after* the emotional response, while cognitive reappraisal is categorized as an “antecedent-focused” emotional regulation strategy, where the emotional stimuli are modified *before* the emotional response [5,7,10]. As a result of this, expressive suppression, which involves persistent self-monitoring and modulating of emotional responses, is known to be more taxing on cognitive resources and results in increased physiological arousal, anxiety, and negative affect [5,11]. In contrast, cognitive reappraisal, which occurs early in the process and does not require constant self-monitoring and self-regulation, is known to be a more adaptive, less taxing strategy that leads to better outcomes, such as neutralizing physiological arousal, decreased anxiety, and negative affect [7,12]. Accordingly, cognitive reappraisal is known to typically result in better mental and physical health outcomes than expressive suppression [5].

Another model, known as the person-by-situation approach, proposes that the utilization of these emotion regulation strategies is based on the context in which they are used [13]. For example, the cognitive reappraisal strategy is said to be more adaptive when the external environment is out of one’s control and the individual can regulate only the self, versus when the external environment presents more controllable stressors. A study conducted with 170 participants in a lab-based setting showed that in the event of uncontrollable stress, higher cognitive reappraisal abilities were associated with reduced depression, whereas in the event of controllable stress, these abilities were associated with increased depression. This study demonstrated the person-by-situation approach wherein certain emotion regulation strategies may not be adaptive or maladaptive per se, but the context influences their adaptability [13]. Thus, the role of emotion regulation strategies and the context in which they are used are worth examining in first responder populations.

Limited research to date has examined the role of emotion regulation in shaping the mental health of first responders. Amongst manifold emotion regulation strategies, expressive suppression and cognitive reappraisal stand out as most pertinent due to their impact on mental health outcomes [5,14,15]. A study that explored the associations between occupational stressors, emotion regulation strategies, and burnout in 602 nurses found that less frequent use of cognitive reappraisal and more frequent use of the expressive suppression in the context of a high level of stressors were associated with higher burnout [16]. Another study that explored emotion regulation in 102 rescue workers from the German Red Cross (EMTs paramedics, medical technicians, and intermediates) found that emotion suppression was associated with greater work-related stress and symptoms [1]. Other studies with a general, non-first responder population found that individuals who reported infrequent and ineffective use of these emotion-regulating strategies showed higher levels of depression, anxiety, and PTSD [5]. Another study conducted across 514 employees in six European countries found that emotional exhaustion coupled with occupational-stress-related burnout negatively impacted employees’ productivity [17]. These studies also highlighted that the capacity to regulate emotions and flexibly apply these emotion regulation strategies reduces psychopathology and increases adaptive functioning [5]. Conversely, a study that investigated cognitive reappraisal and expressive suppression in 489 youth participants found that the increased use of cognitive reappraisal predicted increased levels of positive well-being, and increased use of expressive suppression predicted increased levels of negative well-being [18]. Moreover, a study that looked at PTSD in 93 military veteran patients found that individuals with PTSD who used expressive suppression associated with greater PTSD symptoms, while cognitive reappraisal was associated with fewer PTSD symptoms [19]. Lastly, a study that looked at GAD in 325 youth participants found deficits in emotion regulation such as acceptance of emotions and lack of emotion clarity associated with chronic worry and GAD [20].

Past research thus presents valuable work on emotion regulation strategies and their impact on mental health outcomes, such as PTSD, MD, and GAD, identifying emotion regulation as a central component towards enhancing psychiatric vulnerability. Several studies have also found that emotion regulation processes provide a pathway that links stress exposure to a wide range of psychiatric symptoms, demonstrating that emotion regulation plays a mediating role between stress exposure from past adversities and mental health outcomes [21,22,23]. A study conducted by Raio et al. [24] for example, demonstrated that acute stress markedly impairs the regulation of emotions, which in turn enhances susceptibilities to various psychopathologies [24]. Stress might lead to less adaptive emotion regulation strategies as successful execution of emotion regulation depends on proper executive functioning that involves the prefrontal cortex, which is impaired due to the deleterious impacts of stress [24].There is also evidence that using adaptive emotion regulation strategies might protect against the adverse effects of stress on psychopathology [21]. Although previous research has provided some evidence of emotion regulation as a mediating link between stress and mental health outcomes, to our knowledge, there are close to no studies that have examined the role of emotion regulation as a mediator between occupational stressors and mental health outcomes of PTSD, MD, and GAD in the first responder population.

Understanding whether first responders’ occupational stressors are associated with adverse mental health outcomes by contributing to less adaptive emotion regulation strategies could provide critical information for efforts to boost resilience in this population. Thus, the aim of this study was to investigate whether two emotion regulation strategies, i.e., expressive suppression or cognitive reappraisal, mediated the relationship between occupational stressors and development of PTSD, MD, and GAD in first responders cross-sectionally. The study hypothesis was that occupational stressors would share a significant relationship with—PTSD, MD, and GAD—through an increase in expressive suppression and a decrease in cognitive reappraisal within our first responders’ population.

## 2. Methods

### Participants and Procedures

An aggregate of 987 first responders that included paramedics, firefighters, and EMTs was recruited through professional organizations and social media sites across 50 North and South American states within the USA, Virgin Islands, and Puerto Rico. An online survey link was sent to participants who completed the online informed consent form. The survey was conducted through Qualtrics and took approximately 30 min to be completed. A total of 895 participants (90.7%) fully attempted all measures that were administered to them. Participants’ anonymity was maintained. They were compensated for their participation by entering a raffle draw for USD 100 gift cards that would be given to ten random participants. This study was approved by the Montclair State University’s Institutional Review Board (IRB).

## 3. Measures

### 3.1. First Responder Demographics

Information on participants’ age, gender (male, female, transgender, other), education (high school, bachelor’s, master’s degrees), relationship status (single, married, widowed), employment status (full-time, part-time, etc.), age at job initiation, type of first responder job (paramedics, firefighter, EMS, etc.), type of department the first responders worked in (paid or volunteer), geographic area of work (rural or urban), team size (5000 to 25,000 responders), type of first responder agency (local, private, county, etc.), hours worked on a weekly basis, and the total number of years served as a first responder were collected [25,26].

### 3.2. Occupational Stressors

A 21-item list was created from the Occupational Stressors Scale, the Physician Worklife Survey (PWS) [27], and the Sources of Occupational Stress (SOOS) [28] based on consultation and guidance with experts with more than 10 years of experience being a first responder. This list was used to assess occupational stressors. Participants reported their degree of botheration from the listed occupational stressors on a scale from 0 to 4, with the former being “not at all bothered” and the latter being “extremely bothered.” Then, these items were added together to produce a sum value, with higher scores indicating higher levels of botheration as a result of the work-related stressors. The measures that were used to create the list demonstrated strong psychometric properties [9,27]. Cronbach’s alpha of internal consistency for this measure in this study was 0.90.

### 3.3. Emotion Regulation

A 10-item Emotion Regulation Questionnaire (ERQ) [11] was used to measure the two emotion regulation strategies of expressive suppression and cognitive reappraisal on a scale from 1 (strongly disagree) to 7 (strongly agree). Four items assessed expressive suppression, and six items assessed cognitive reappraisal. For expressive suppression, participants were asked to rate items such as “I keep my emotions to myself,” and “I control my emotions by not expressing them.” For cognitive reappraisal, participants were asked to rate items such as “When I want to feel less negative emotion (such as sadness or anger), I change what I’m thinking about.” The scores were summed and used as indicators for each of the two emotion regulation strategies, with higher scores indicating greater levels of each of the two emotion regulation strategies. The ERQ-10 has evidence of strong psychometric properties [29]. Cronbach’s alpha of internal consistency in the current study was 0.76 for expressive suppression and 0.81 for cognitive reappraisal.

### 3.4. PTSD Symptoms

A 20-item Posttraumatic Checklist for DSM-5 (PCL-5) [30] was administered to obtain participants’ information about PTSD symptoms. Participants were asked to report how much various symptoms of PTSD had bothered them in the past month on a scale of 0–5, with the former being “not at all” and the latter being “extremely”. Four areas of PTSD symptoms in accordance with the DSM-5 criteria were assessed that included intrusion, avoidance symptoms, negative alterations in thoughts and mood, and changes in arousal to external stimulus. Items were summed for a total score, with higher scores indicating more severe PTSD symptoms. The PCL-5 has demonstrated strong psychometric properties [31,32]. Cronbach’s alpha for this measure in the current study was 0.96.

### 3.5. MD Symptoms

A 9-item Patient Health Questionnaire (PHQ-9) [33] was used to assess depression symptoms on a scale of 0–4, with the former being “not at all” and the latter being “nearly every day.” Items were summed for a total score, with higher scores indicating more severe depressive symptoms. The PHQ-9 has demonstrated strong psychometric properties [34,35]. Cronbach’s alpha of this measure in this study was 0.92.

### 3.6. GAD Symptoms

A 7-item Generalized Anxiety Disorder scale (GAD-7) [36] was used to assess GAD symptoms on a scale of 0–4, with the former being “not at all” and the latter being “nearly every day.” Items were summed for a total score, with higher scores indicating more severe GAD symptoms. The GAD-7 has demonstrated strong psychometric properties [37]. Cronbach’s alpha of this measure in this study was 0.94.

## 4. Data Analyses

The analyses consisted of various steps. First, preliminary tabulations were done to obtain descriptives and bivariate correlations of the main variables of the study. Second, six mediation models were run while simultaneously controlling for potentially confounding variables, including first responder demographics such as employment status, type of first responder job, type of first responder department, geographic location, and type of first responder agency. The models examined the association between occupational stressors and PTSD, MD, and GAD symptoms via hypothesized mediators of emotional regulation, expressive suppression and cognitive reappraisal. Third, indirect effects were examined in these linear models using 95% bias corrected confidence intervals that used 5000 bootstrapped samples. All models were run using Model 4 PROCESS macro 3.0 [38] and all data analyses were done using IBM SPSS Statistics 25 [39].

## 5. Results

### 5.1. Descriptive Statistics

The sample consisted of participants who showed an age range of 18–73 years (M = 37.32, SD = 12.09), with a higher percentage of male gender (59.2%). Please see Table 1 for a detailed visual illustration of first responder demographics. Table 2 depicts the mean values and standard deviations for key study variables, as well as bivariate correlations among them. Bivariate correlation analysis revealed that occupational stressors were positively correlated with expressive suppression and each mental health outcome but not significantly correlated with cognitive reappraisal. Furthermore, expressive suppression was positively correlated with each mental health outcome, whereas cognitive reappraisal was negatively correlated with each outcome.

### 5.2. Inferential Statistics

#### Expressive Suppression as a Mediator

**PTSD symptoms.**Figure 1a illustrates findings from the occupational stressors model and PTSD symptoms. It was found that first responders who reported a greater number of occupational stressors also showed significant positive associations with expressive suppression (*B* = 0.07, *SE* = 0.01, *p* < 0.001). Participants who experienced more occupational stressors also reported significantly greater PTSD symptoms (*B* = 0.62, *SE* = 0.03, *p* < 0.001) with the inclusion of expressive suppression in the model. Higher expressive suppression was associated with significantly higher PTSD symptoms (*B* = 0.77, *SE* = 0.10, *p* < 0.001). The 95% confidence interval for the indirect effect of occupational stressors on PTSD symptoms via expressive suppression did not include zero, indicating statistical significance (*B* = 0.05; 95% CI: 0.03, 0.08).

**MD symptoms.**Figure 1b illustrates findings from the occupational stressors model and MD symptoms. It was found that first responders who reported a greater number of occupational stressors also showed significant positive associations with expressive suppression (*B* = 0.07, *SE =* 0.01, *p* < 0.001). Participants who experienced more occupational stressors also reported significantly greater MD symptoms (*B* = 0.23, *SE =* 0.01, *p* < 0.001) with the inclusion of expressive suppression in the model. Higher expressive suppression was associated with significantly higher MD symptoms (*B* = 0.30, *SE =* 0.04, *p* < 0.001). The 95% confidence interval for the indirect effect of occupational stressors on MD symptoms via expressive suppression did not include zero, indicating statistical significance (*B* = 0.02, 95% CI: 0.01, 0.03).

**GAD symptoms.** Figure 1c illustrates findings from the occupational stressors model and GAD symptoms. It was found that first responders who reported a greater number of occupational stressors also showed significant positive associations with expressive suppression (*B* = 0.07, *SE* = 0.01, *p* < 0.001). Participants who experienced more occupational stressors also reported significantly greater GAD symptoms (*B* = 0.20, *SE* = 0.01, *p* < 0.001) with the inclusion of ES in the model. Higher expressive suppression was associated with significantly higher GAD symptoms (*B* = 0.17, *SE* = 0.03, *p* < 0.001). The indirect effect of occupational stressors on GAD symptoms via expressive suppression was significant (*B* = 0.01, 95% CI: 0.01, 0.02).

### 5.3. Cognitive Reappraisal as a Mediator

The results of models including cognitive reappraisal as a mediator are shown in Figure 2a–c. No indirect effect was found between occupational stressors and PTSD, MD, or GAD symptoms through cognitive reappraisal.

## 6. Discussion

The cross-sectional study examined whether occupational stressors were indirectly associated with the three mental health outcomes (PTSD, MD, and GAD symptoms) through the two emotion regulation strategies, expressive suppression, and cognitive reappraisal, in a sample of first responders. Higher occupational stressors and expressive suppression were positively associated with PTSD, MD, and GAD symptoms, whereas cognitive reappraisal was negatively associated with PTSD, MD, and GAD symptoms. In addition, whereas the indirect pathways from occupational stressors to the PTSD, MD, and GAD symptoms via expressive suppression were positive and significant, the indirect pathways via cognitive reappraisal were non-significant.

The findings in our study elucidate that first responders’ risk of increased PTSD, MD, and GAD symptoms was not just the result of the occupationally related stressor that they were facing but also due to the active use of the expressive suppression strategy of emotion regulation. As previously stated, according to the Gross’s process model of emotion regulation, expressive suppression is a response-focused strategy, one that is used during or after the event [5,7]. Our findings suggest that when first responders chose (whether conscious or unconscious) to use a response-focused strategy to deal with occupational stressors, their symptoms of PTSD, MD, and GAD were enhanced. Our results buttress previous findings that higher occupational stress and higher expressive suppression predicted higher levels of PTSD, MD, and GAD symptoms [1,4,9,16,18,19,20]. Our study thus provides additional evidence that using the emotion regulation strategy of expressive suppression could negatively impact mental health outcomes, specifically in the first responder population. It is also important to note that our study revealed an indirect mechanistic pathway, showing that increased expressive suppression acted as a mediator, enhancing the negative effects of occupational stressors on first responders’ mental health.

Cognitive reappraisal was significantly negatively associated with our first responders’ mental health outcomes, suggesting that cognitive reappraisal strategy was an adaptive strategy that may have helped ameliorate PTSD, MD, and GAD symptoms in this population. However, there was no indirect pathway found from occupational stressors to these mental health outcomes via cognitive reappraisal, which supports the detrimental influence of the response-focused strategy of expressive suppression versus an antecedent-focused strategy of cognitive reappraisal [5,7]. The finding also indicates that occupational stressors may contribute to expressive suppression, but not cognitive reappraisal. Previous studies have shown that trauma-exposed individuals with greater perceived stress show decreased tendency to use cognitive reappraisal strategies [8,40]. A possible explanation is that acute stress, such as occupational stress, has been found to impair the functioning of the prefrontal neural systems, such as the inhibition of the prefrontal cortex and simultaneous activation of amygdala responses, resulting in less effectiveness of the use of cognitive reappraisal strategies under stress. Executive functions, such as cognitive flexibility, are found to be impaired under very stressful situations [8,40,41]. Another possible explanation, based on anecdotal evidence [42], may be that first responder occupational context might pull for workers to suppress their outward expressions of emotions, thereby increasing expressive suppression after a stressful event, either because of workplace norms or the requirements of the job—i.e., to maintain a calm demeanor and continue one’s duties in the face of stress [42,43].

Taken together, this pattern of results suggests the need to train and educate first responders in antecedent focused strategies such as cognitive reappraisal that have shown to reap beneficial effects on mental health outcomes in previous studies [7,12,16,18,19]. Simultaneously, first responders must also be educated about recognizing and understanding the drawbacks of using expressive suppression for their mental health.

Another possible explanation for the impact of expressive suppression versus cognitive reappraisal in our first responder population could have been due to the context influencing the emotion regulation strategy, in accordance with the previously mentioned person-by-situation approach model [13]. Based on this model, perhaps understanding which emotion regulation strategy first responders used more than the other within the context of controllable versus uncontrollable stressors from the external environment would have helped determine whether the event of uncontrollable stress could have triggered the use of higher cognitive reappraisal abilities before the event versus utilization of expressive suppression strategy after the event. Although we were unable to explore this possibility with our data, this could be a useful direction for future research.

## 7. Implications

Our findings suggest various implications for first responder populations, which not only include the importance of decreasing the utilization of expressive suppression as an emotion regulation strategy in response to occupational stressors, but a greater proactiveness, education, training, and preparation in antecedent-focused strategies such as cognitive reappraisal as well as the ability to understand how one’s context influences the chosen strategy is required. First responders’ use of expressive suppression to manage the experience of occupational stress likely exhausts cognitive resources by requiring a continuous effort to inhibit all negative feelings arising from the occupational stress. Results support the need for workplace interventions that aim to protect first responders from developing psychiatric symptoms by encouraging effective processing of occupational stress through cognitive reappraisal strategies versus expressive suppression. Teaching first responders to use cognitive reappraisal may diminish the burden of occupational stress by altering the inherent negative meaning.

Our results also support the value of programs that aim to facilitate or boost the utilization of more adaptive emotion regulation strategies in first responder populations. A study that investigated the effectiveness of emotion regulation training in critical care nurses found that eight sessions of training for a duration of three weeks effectively reduced occupational stress in this population, thereby mitigating potential adverse mental health outcomes [44]. The training utilized in the study involved educating about emotions, their short-term and long-term effects, the behavioral and physiological outcome of emotions, teaching interpersonal skills such as effective communication, emotion expression, problem solving or conflict resolution, practicing expanding attention, reduction of mental rumination, and learning ways to eliminate barriers to implementing these skills [44]. It would be interesting to see the effect of this type of intervention in our first responder population.

## 8. Limitations

Our study had several key limitations that could be addressed in future research. First, using a non-representative sample limited the generalizability of our findings, and only first responders within the United States were included. As such, future research must incorporate replication to investigate whether the pattern of these findings might also hold within the general population of first responders and among first responders in other countries. Second, since this was a cross-sectional study, the findings are limited to their correlational pattern and cannot be considered causal unless a longitudinal, prospective study is done in the future to validate these findings. Third, the use of self-report measures does not provide accurate clinical diagnostic claims for participants who reported their levels of PTSD, MD, and GAD symptoms. Future research must also consider clinically diagnosing these mental health outcomes to complement the self-report measures. Finally, the use of our occupational stressor scale that utilized a 21-item list drawn from other measures has only been piloted in this study, with future studies needed in this area to validate this list’s psychometric properties per se or replicate this study’s results using a more established measure. For future study, it is also worth exploring what type of emotion regulation strategies (expressive suppression or cognitive reappraisal) were used predominantly by the first responders that could help provide stronger evidence for the person-by-situation approach model.

Despite these limitations, this study provides preliminary evidence for the negative impact of occupational stressors on first responders’ mental health through the increased use of expressive suppression as an emotion regulation strategy. The results suggest the importance of practices that bolster first responders’ ability to use more effective and adaptive emotion regulation strategies such as cognitive reappraisal that might help enhance psychological resilience.

## 9. Conclusions

In conclusion, this study aimed to investigate the mediating role of two emotion regulation strategies, expressive suppression or cognitive reappraisal, between first responders’ occupational stressors and three mental health outcomes, PTSD, MD, and GAD symptoms, using cross-sectional data and found that expressive suppression was a significant mediator between occupational stressors and PTSD, MD, and GAD symptoms, but not cognitive reappraisal.

## Figures and Tables

**Figure 1 ijerph-19-07009-f001:**
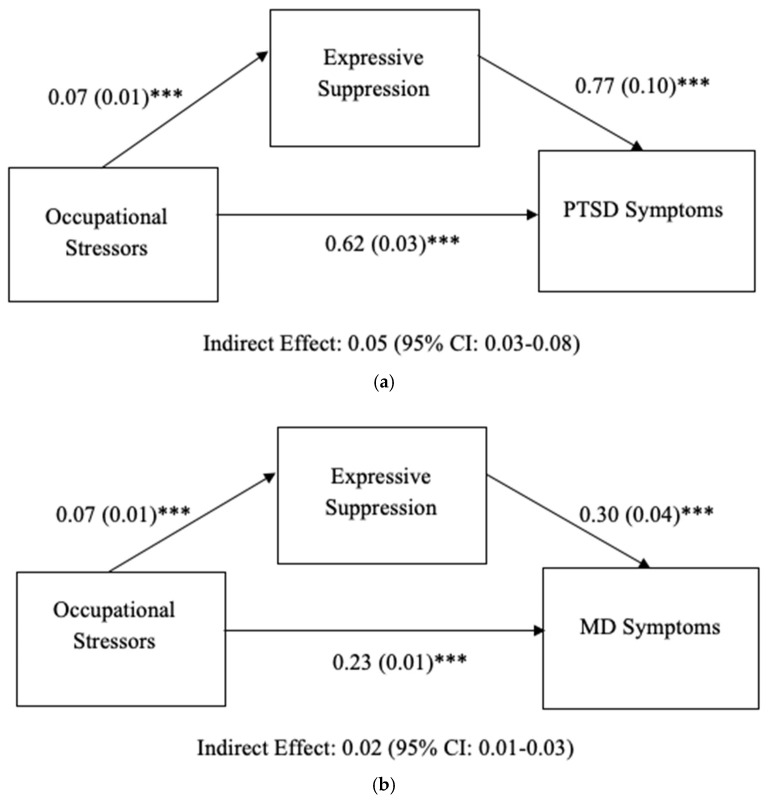
(**a**) Results of mediation model predicting PTSD. Analysis controlled for demographic (age, gender) and work-related (first response type, department, area, employment status, agency) variables. Occupational Stressors, Expressive Suppression, PTSD: Posttraumatic Stress Disorder. (**b**) Results of mediation model predicting MD. Analysis controlled for demographic (age, gender) and work-related (first response type, department, area, employment status, agency) variables. Occupational Stressors, Expressive Suppression, MD = Major Depression. (**c**) Results of mediation model predicting GAD. Analysis controlled for demographic (age, gender) and work-related (first response type, department, area, employment status, agency) variables. Occupational Stressors, Expressive Suppression, GAD: Generalized Anxiety Disorder. Unstandardized regression coefficients (*B*) are listed. *** *p* < 0.001.

**Figure 2 ijerph-19-07009-f002:**
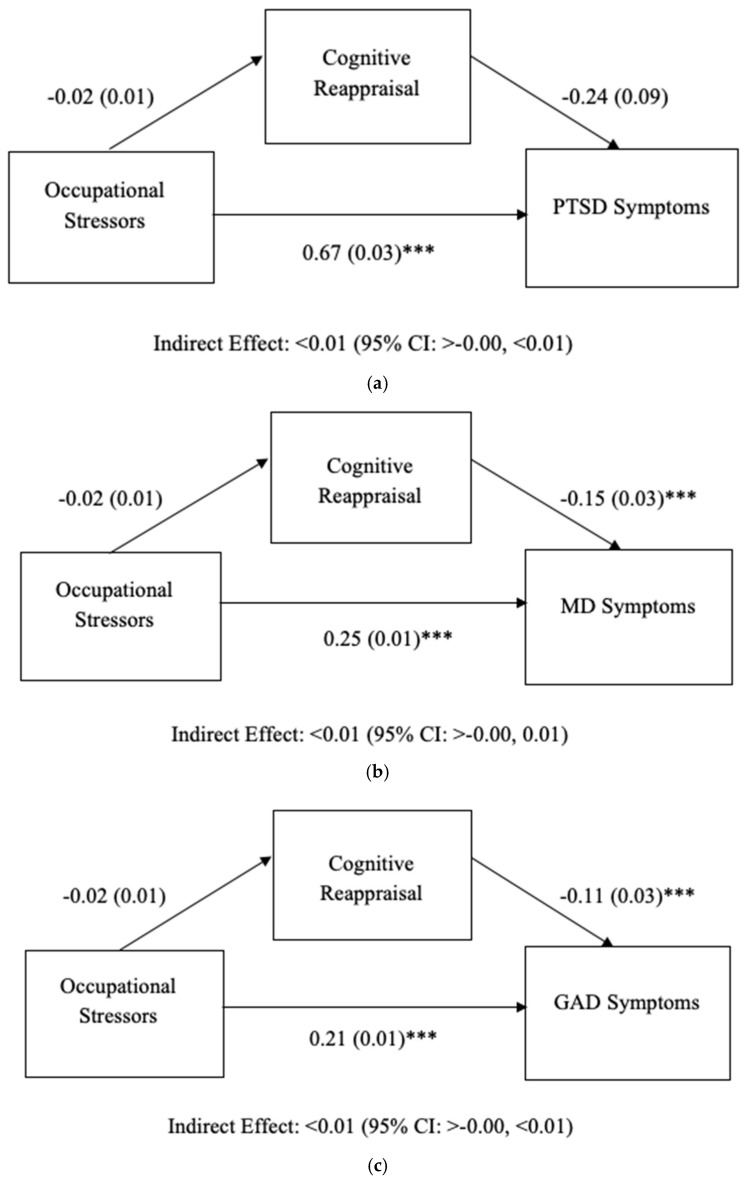
(**a**) Results of mediation model predicting PTSD. Analysis controlled for demographic (age, gender) and work-related (first response type, department, area, employment status, agency) variables. Occupational Stressors, Cognitive Reappraisal, PTSD: Posttraumatic Stress Disorder. (**b**) Results of mediation model predicting MD. Analysis controlled for demographic (age, gender) and work-related (first response type, department, area, employment status, agency) variables. Occupational Stressors, Cognitive Reappraisal, MD: Major Depression. (**c**) Results of mediation model predicting GAD. Analysis controlled for demographic (age, gender) and work-related (first response type, department, area, employment status, agency) variables. Occupational Stressors, Cognitive Reappraisal, GAD: Generalized Anxiety Disorder. Unstandardized regression coefficients (*B*) are listed. *** *p* < 0.001.

**Table 1 ijerph-19-07009-t001:** Participant demographics and work-related covariates (*N* = 895).

	*N*	%	*M (SD)*
*Demographics:*			
Age (years)			37.32 (12.09)
Male gender			
Yes	530	59.2	
No	365	40.8	
*FR work-related characteristics:*			
Age at start of FR (years)			21.22 (6.91)
*FR work type:*			
Fire only	52	5.8	
EMS only	534	59.7	
Both—combined	175	19.6	
Both—separate agencies	134	15.0	
*FR department type:*			
Volunteer	146	16.3	
Career	525	58.7	
Combined	224	25.0	
*Response area type:*			
Rural	219	24.5	
Suburban	337	37.7	
Urban	339	37.9	
*Employment status:*			
Full time	611	68.3	
Part-time	79	8.8	
Per diem	62	6.9	
Volunteer	143	16.0	
*Time as a FR:*			
Less than 1 year	11	1.2	
1–5 years	179	20.0	
6–10 years	187	20.9	
11–15 years	163	18.2	
16–20 years	112	12.5	
21–25 years	78	8.7	
More than 25 years	165	18.4	
*FR agency type:*			
Federal	10	1.1	
State	30	3.4	
County	161	18.0	
Local	358	40.0	
Private	284	31.7	
Other	52	5.8	

**Table 2 ijerph-19-07009-t002:** Means, standard deviations, and bivariate correlations between key study variables (*N* = 895).

	1	2	3	4	5	6
1. Occupational stressors	—					
2. Expressive suppression	0.22 ***	—				
3. Cognitive reappraisal	−0.06	0.03	—			
4. PTSD symptoms	0.60 ***	0.33 ***	−0.11 **	—		
5. MD symptoms	0.59 ***	0.31 ***	−0.15 ***	0.79 ***	—	
6. GAD symptoms	0.58 ***	0.24 ***	−0.13 ***	0.77 ***	0.83 ***	—
M	37.50	17.78	29.96	25.86	9.56	8.57
SD	15.85	5.12	5.73	18.59	7.20	6.34

Note. ** *p* < 0.01; *** *p* < 0.001.

## Data Availability

The data presented in this study are available on request from the corresponding author.

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
