# Peer review of "Investigating Strategies of Emotion Regulation As Mediators of Occupational Stressors and Mental Health Outcomes in First Responders"

_ijerph, 2022, doi:10.3390/ijerph19127009_

Round 1

Reviewer 1 Report

 I suggest the abstract could be more readable for audience. Last sentence - What could be more effective and adaptive emotion regulation strategies for the them?

Author Response

We thank Reviewer 1 for this helpful comment. We revised the final sentence of our abstract to specify the adaptive emotion regulation strategies, so that it reads: 

"Even though these findings are in purview of a cross-sectional research design, they suggest the importance of practices that bolster first responders’ ability to use more effective and adaptive emotion regulation strategies such as emotion expression, effective communication, and cognitive reappraisal that might help enhance psychological resilience."

Reviewer 2 Report

This manuscript is very complete in terms of research methods and content, the only thing that needs to be revised is the format of the literature citations. The format used in the manuscript is APA, which differs from the requirements of MDPI. It is recommended to modify the bibliographic citation format.

Author Response

We thank Reviewer 2 for this helpful comment and the attention to detail. Since the IJERPH Instructions for Authors page (https://www.mdpi.com/journal/ijerph/instructions) stated the following: 

"Your references may be in any style, provided that you use the consistent formatting throughout. It is essential to include author(s) name(s), journal or book title, article or chapter title (where required), year of publication, volume and issue (where appropriate) and pagination. DOI numbers (Digital Object Identifier) are not mandatory but highly encouraged. The bibliography software package EndNoteZoteroMendeleyReference Manager are recommended."

We would like to stick to the APA format consistently used throughout the manuscript as this does not violate the IJERPH recommendation. Thank you! 

Reviewer 3 Report

Dear Authors, 

Your work is interesting and timely.

Overall the literature review focus the key thematic of the article, but it would be helpful for the authors to go through it again with the intention of being more precise and documenting with more recent and on going literature. It could be improved with more up to date works. For instance, you can take a look at this article: https://doi.org/10.3390/ijerph18052425

It would also be useful to present recommendations for future avenues of research.

Author Response

We thank Reviewer 3 for these helpful comments.

  • Minor spell check completed.
  • Our introduction consists of up-to-date articles that precisely address our variables of interest. We also added the recommended reference provided by the reviewer so that it reads (page 3): 

"Another study conducted across 514 employees in six European countries found that emotional exhaustion coupled with occupational-stress-related burnout negatively impacted employees’ productivity (Leitão et al., 2021). "

  • We already presented recommendations for future avenues of research, within our limitations section of the manuscript and made this clearer, so that it reads (page 8): 

"Our study had several key limitations that could be addressed in future research. First, using a non-representative sample limited the generalizability of our findings, and only first responders within the United States were included. As such, future research must incorporate replication to investigate whether the pattern of these findings might also hold within the general population of first responders and among first responders in other countries. Second, since this was a cross-sectional study, the findings are limited to their correlational pattern and cannot be considered causal unless a longitudinal, prospective study is done in the future to validate these findings. Third, the use of self-report measures does not provide accurate clinical diagnostic claims for participants who reported their levels of PTSD, MD, and GAD symptoms. Future research must also consider clinically diagnosing these mental health outcomes to complement the self-report measures. Finally, the use of our occupational stressor scale that utilized a 21-item list drawn from other measures has only been piloted in this study, with future studies needed in this area to validate this list’s psychometric properties per se, or replicate this study’s results using a more established measure. For future study, it is also worth exploring what type of emotion regulation strategies (expressive suppression or cognitive reappraisal) were used predominantly by the first responders that could help provide stronger evidence towards the person-by-situation approach model.

Despite these limitations, this study provides preliminary evidence for the negative impact of occupational stressors on first responders’ mental health through the increased use of expressive suppression as an emotion regulation strategy. The results suggest the importance of practices that bolster first responders’ ability to use more effective and adaptive emotion regulation strategies such as cognitive reappraisal that might help enhance psychological resilience."

Reviewer 4 Report

The paper analyses the mediational role of emotional suppression and reappraisal in the relationship between work stressors and different measures of occupational health. 

The paper has several key limitations.

The first is its lack of novelty. The mediating relationship of emotional regulation strategies has been extensively studied in the field of occupational health. A major part of this research has been conducted over three decades under the category of emotional labour. The authors do not include all of this evidence in their study. This partial review of the literature constitutes a second important limitation.

Thirdly, as this is a widely researched topic, the authors are expected to make their hypotheses explicit in the introduction (i. e. H1.....).

The fourth limitation has to do with the design. The use of cross-sectional data to test mediational models is highly inadvisable. Its use might be justified in an exploratory study on a topic for which there is little or no previous evidence. In this case, authors should make the effort to collect longitudinal data in order to offer results that make some contribution. 

Fifth, the analysis strategy should be more parsimonious. From a statistical point of view, it is advisable to include both proposed mediators simultaneously. This will reduce the number of Figures from 6 to 3.

Sixth, the inclusion of control variables must be theoretically justified. Why do you introduce these variables? How are they related to the variables in the model? If there is no justification, they introduce unnecessary "noise" into the model,

Seventh, the explanations they offer for the absence of significant spillover effects in the case of reappraisal are dubiously valid. 

For example, I do not understand the logic of this statement "However, there was no indirect pathway found from occupational stressors 300 to these mental health outcomes via cognitive reappraisal, which supports the detrimental 301 influence of the response-focused strategy of expressive suppression versus an antecedent 302 focused strategy of cognitive reappraisal".

The second explanation for the non-significance of the indirect effects of reappraisal would explain a negative relationship between stressors and reappraisal, not the non significant correlation the authors founded.

This other explanation is not clear "thereby increasing expressive suppression after a stressful event, either because of workplace norms or the requirements of the job - i.e., to 316 maintain a calm demeanor and continue one's duties in the face of stress". Non-expression  of emotions can be achieved with reappraisal.

Overall, the explanations are highly speculative and not always correct.

Taking all this elements into consideration, I´m recomending the paper to be rejected for publication. Authors should add further studies that contribute step further the knowledge on the field.

Author Response

We thank Reviewer 4 for the helpful comments and the feedback.

  • We have delineated the findings of our research in light of the limitations of our research design, while also suggesting the need for a future, longitudinal study design so that it reads (page 8): 

"Second, since this was a cross-sectional study, the findings are limited to their correlational pattern and cannot be considered causal unless a longitudinal, prospective study is done in the future to validate these findings." 

  • We clearly and explicitly stated our hypothesis within our introduction section at the end, so that it reads (page 3).

"The study hypothesis was that occupational stressors would share a significant relationship with –– PTSD, MD, and GAD –– through an increase in expressive suppression and decrease in cognitive reappraisal, within our first responders’ population."

  • We theoretically justified the use of our control variables, so that it reads (page 5): 

"Second, six mediation models were run, while simultaneously controlling for potentially confounding variables including, first responder demographics such as employment status, type of first responder job, type of first responder department, geographic location, and type of first responder agency. 

Round 2

Reviewer 4 Report

Thanks to the authors for the improvements made to the paper. However, in order to appreciate its dimension, it is necessary that the authors follow the usual format of response to a reviewer. That is, include the literal text of each of my comments, followed by the form in which it has been addressed.

I anticipate that the paper presents important limitations derived from the chosen design that are difficult to overcome and that make it very likely that my recommendation for the paper will continue to be rejected for publication.